# Microbial assemblages and methanogenesis pathways impact methane production and foaming in manure deep-pit storages

Fan Yang[1]*, Daniel S. Andersen[1], Steven Trabue[2], Angela D. Kent[3], Laura M. Pepple[3]¤, Richard S. Gates[4], Adina S. Howe[1]

1 Department of Agricultural and Biosystems Engineering, Iowa State University, Ames, Iowa, United States of America, 2 USDA-Agricultural Research Service, National Laboratory for Agriculture and the Environment, Ames, Iowa, United States of America, 3 The Department of Natural Resources and Environmental Sciences, University of Illinois at Urbana-Champaign, Urbana, Illinois, United States of America, 4 Egg Industry Center, Iowa State University, Ames, Iowa, United States of America

¤ Current address: Puck Custom Enterprises, Manning, Iowa, United States of America
* fan.michelle.yang@gmail.com

**Data Availability Statement:** All relevant data are within the paper and its Supporting Information files.

## Abstract

Foam accumulation in swine manure deep-pits has been linked to explosions and flash fires that pose devastating threats to humans and livestock. It is clear that methane accumulation within these pits is the fuel for the fire; it is not understood what microbial drivers cause the accumulation and stabilization of methane. Here, we conducted a 13-month field study to survey the physical, chemical, and biological changes of pit-manure across 46 farms in Iowa. Our results showed that an increased methane production rate was associated with less digestible feed ingredients, suggesting that diet influences the storage pit's microbiome. Targeted sequencing of the bacterial 16S rRNA and archaeal *mcrA* genes was used to identify microbial communities' role and influence. We found that microbial communities in foaming and non-foaming manure were significantly different, and that the bacterial communities of foaming manure were more stable than those of non-foaming manure. Foaming manure methanogen communities were enriched with uncharacterized methanogens whose presence strongly correlated with high methane production rates. We also observed strong correlations between feed ration, manure characteristics, and the relative abundance of specific taxa, suggesting that manure foaming is linked to microbial community assemblage driven by efficient free long-chain fatty acid degradation by hydrogenotrophic methanogenesis.

## Introduction

Animal production has shifted from pasture systems to confinement facilities as larger, more specialized operations replace smaller less efficient farms to meet inexpensive protein demands [1]. These shifts have resulted in the separation of farrowing and finishing operations, implementation of liquid manure storage systems, and greater use of concentrates in animal rations

**Funding:** This project was funded by National Pork Board and Iowa Pork Producers Association (project number: 15-136). The funders had no role in study design, data collection and analysis, decision to publish, or preparation of the manuscript.

**Competing interests:** The authors have declared that no competing interests exist.

for swine production. In the Midwestern United States, manure is predominately stored underground in 2.4–3 m deep pits for extended periods (6–12 months). Deep-pits are built within the swine production facility below a slatted floor housing animals to reduce nutrients loss and dilution [2].

Swine growers using deep-pit manure management systems have observed sporadic foam formation. Farms experiencing manure foaming face significant management issues as manure foam limits facility storage space and requires more frequent removal. More importantly, the foam traps methane and other gases resulting in potentially life-threatening fires and explosions [2]. In the Midwestern US, there has been an increase in the frequency of foaming events [3, 4]. The inability to replicate this phenomenon in the laboratory makes ascertaining the cause of foaming a challenge [5, 6]. Consequently, the only knowledge of its causes is mainly limited to its correlation with high ($\geq$ 0.1 L methane / L manure • day) methane production rates (MPR), with little knowledge of its abiotic or biotic drivers. In this study, we performed a large-scale characterization of over 500 manure samples collected monthly for 13 months across 46 swine farms in Iowa to understand the dietary and microbial associations that drive manure physical-chemical change.

## Materials and methods

### Experimental design

Over 500 manure samples were collected from 46 farms deep pits (8ft) within Iowa over thirteen months (S1 Fig). The farms worked with two integrators and received feed from four feed mills (S1 Table). Data were collected to describe each farm's feed ingredients. Manure sampling procedures and characterization (e.g., total solids, methane production rate) have previously been described [7]. Additional manure characterization (i.e., pH, moisture, organic N) was performed by Midwest Laboratories (Omaha, NE). At collection, the manure storage surface was characterized as non-foaming (no-foam, with direct access to liquids), crust-forming (crust, with a hard and dry layer on top), or foaming (foam, with visible bubbles) (S2 Fig).

### DNA extraction and sequencing

DNA was extracted from samples obtained from the top layer of non-foaming pits or the transition layer below the foam or crust in other pits (S3 Fig, layer B). Additionally, a subset of manure slurry samples was used to characterize methanogen populations (S3 Fig, layer C). The genomic DNA was extracted from 200 mg manure samples using the FastDNA SPIN Kit for Soil (MP Biomedical).

Extracted DNA was sequenced to characterize bacterial and methanogen communities. To identify bacteria in manure samples, the V4-V5 region of the 16S rRNA gene was amplified with primers 515F 5'-GTGCCAGCMGCCGCGGTAA-3' and 924R 5'-CCGTCAATTCMTT-TRAGT-3' with barcodes and Illumina adaptors added as previously described [8]. The methane-production gene, methyl-coenzyme A reductase (*mcrA*), was used to identify methanogens in manure samples. The *mcrA* gene was amplified using barcode and Illumina adaptor added primers mlas 5'-GGTGGTGTMGGDTTCACMCARTA-3' and mcrA-rev 5'-CGTTCATBGCGTAGTTVGGRTAGT-3' [9]. Every 50 ul PCR reaction contained 25uL 2X KAPA HiFi HotStart ReadyMix (KAPA Biosystems, Woburn, MA, USA), 500 uM each primer, 50 ng template DNA, and 21uL DNA-free water. Thermal cycling conditions for this reaction included an initial denaturation at 98°C for 45 sec., 30 cycles of 98°C for 10 sec., 55°C for 30 sec., 72°C for 30 sec., followed by a final extension at 72°C for 2 min. The first 15 cycles had a temperature ramp rate at 0.6°C/s, and the next 15 cycles had a temperature ramp rate at

3˚C/s. The primer-dimers were removed using 0.8 X volume of AMPure[®] XP beads (Agencourt Bioscience, Beverly, MA, USA).

For sequencing library preparation, an equal amount of amplicons from each sample were pooled together. The pooled samples were sent to Roy J. Carver Biotechnology Center (Urbana, IL, USA) for sequencing on an Illumina MiSeq instrument with a 2 x 250 bp reads configuration using Nano Kit v2 (Illumina, San Diego, CA, USA). All sequences are deposited in National Center for Biotechnology Information (NCBI) Sequence Read Archive (SRA) with accession numbers SRR5564278—SRR5564520 and SRR5566243—SRR5566590.

### DNA sequence processing

Pair-ended bacterial 16S rRNA gene sequences were assembled using the Ribosomal Database Project (RDP) Paired-end Reads Assembler[7] with minimal overlap of 80 bases (-o 80) and minimal assembled length 350 bases (–l 350). Assembled sequences with an expected maximum error-adjusted Q score less than 25 over the entire sequence were eliminated. Usearch (8.1, 64bit) [10] was used to remove chimeras de novo, followed by removing chimeras of known reference genes using the RDP 16S rRNA gene training set sequences (No. 15). High quality and chimera-filtered sequences were clustered at 97% sequence similarity by CD-HIT (4.6.1) [11], resulting in identifying unique operational taxonomic units (OTUs) and their abundance in each sample. CD-HIT was used because of its speed and previously demonstrated production of clusters highly similar to the actual number of OTUs from simulated complex data [12, 13]. Each representative OTU sequence's taxonomy was identified based on RDP 16S rRNA database using RDP Classifier [14] with a confidence cutoff at 50% (-c 0.5). At least 98.26% of the OTUs could be identified at the bacterial phylum level. To preserve the microbial community composition and avoid additional biases, we opted to remove questionable OTUs and inadequately sequenced samples and use raw counts and relative abundance for downstream analyses [15, 16]. Specifically, OTUs that were observed fewer than five times across all samples were removed, and samples with less than 10,000 sequence reads ($< 0.97$ Good's coverage index) were also excluded from the analysis, resulting in a total of 488 samples used for analyses (S2 Table).

The methanogen-associated *mcrA* gene sequences were processed similarly to bacterial 16S rRNA gene sequences with the following modifications. The assembled *mcrA* gene sequence length was restricted between 400 and 460 bases. After de novo chimera removal, a non-redundant version of the previously published *mcrA* database was used as the reference dataset to remove any chimeras that were missed by de novo methods [17]. The dataset was then used to construct a Basic Local Alignment Search Tool (BLAST) database to identify the methanogen OTU taxonomy using BLAST+ (2.2.30) [18]. The sequences matched were all significant (with a maximum expected error of 1E-78, and a minimum percent identity of 79%). Similar to the bacterial sequences, methanogen OTUs observed fewer than five times across all samples were removed, and samples with less than 4,000 sequences ($< 0.99$ Good's coverage index) were excluded from the analysis (S2 Table).

### Statistical analyses

Diet information and manure characteristics were analyzed for significant correlations with foam, crust, and no-foam manure using Bayes factor analysis [19]. This analysis was performed in R (3.2.4) using the package BayesFactor (0.9.12–2) [20]. Correlations were estimated between dietary inputs and manure characteristics. Monotonic relationships were evaluated using Spearman's correlation analysis, and non-monotonic relationships were evaluated using the Hoeffding dependence test [21]. The significance of each relationship (p-value) was

adjusted for false discovery rate using the method developed by Benjamini & Hochberg [22]. If the adjusted p-value of Spearman's correlation was less than 0.05, the relationship was considered monotonic, and Spearman's correlation coefficient (ρ) was used to determine how strong the correlation was. If the adjusted p-value of the Hoeffding dependence test was less than 0.05, but the adjusted p-value of Spearman's correlation analysis was greater or equal to 0.05, the relationship was considered non-monotonic. The Hoeffding dependence coefficient (D) was used to determine the strength of the correlation. The correlation analysis was performed in R (3.2.4) using packages Hmisc (3.17–3).

Similarities between bacterial communities identified in foam, no-foam, and crust samples were evaluated. To distinguish the most critical factors contributing to the bacterial community variations, we standardized the OTU abundance across all samples by the total number of sequences per sample. The Bray-Curtis distance was calculated to evaluate the community dissimilarities among samples. Permutational multivariate analysis (PERMANOVA) was used to independently test if diet ingredients and manure physical and chemical characteristics impacted microbial community variance between foaming and non-foaming storages. The community ordinations were performed in R (3.2.4) using packages Vegan (2.3–5). Additional descriptions of the statistical analyses are in the S1 File.

## Results and discussion

### Manure surface textures

Manure samples were visually categorized based on surface texture into three types: 1) foam, 2) no-foam, and 3) crust. Manure surface textures varied greatly from farm-to-farm and within a farm throughout the 13 months (S1 Fig). No distinct patterns of manure surface texture over time were identified within individual pits. Overall, we observed that 40–60% of pit samples collected each month were foam samples. Additionally, the proportion of no-foam samples collected decreased from over 50% in 2012 to less than 30% in 2013. The proportion of samples with crusts increased from less than 8% to above 32% from 2012 to 2013 (Fig 1). Together, these trends suggest that manure foaming was persistent, and long-term stored manures were likely to form crusts or foam during the time of this study.

### Specific feed ingredients were observed to influence manure foaming

The composition of swine feed was observed to influence characteristics of the manure surface texture in pits. No-foam manures were associated with more digestible feed, and foam and crust manure associated with the less digestible feed. In particular, soybean meal (SBM) levels were significantly higher in diets associated with no-foam manure, and the proportion of distiller's dried grains with soluble (DDGS) was significantly higher in feed given to swine from foam and crust manures (Table 1). SBM is more digestible than DDGS due primarily to DDGS formulated diets having higher neutral detergent fiber (NDF) contents than SBM-based diets [23–25]. Lower digestibility of DDGS diets increases the amount of fecal output and significantly increases manure foaming potential (Table 1) [4, 24, 26, 27]. Increasing poorly digestible diet components, including NDF, can induce an anti-nutritive effect in swine [28]. While neutral detergent fiber has lower digestibility by pigs, the excreted partially degraded NDF particles are rich in plant polysaccharides. These polysaccharides are highly digestible by anaerobic microbial communities, which support manure fermentation in the storage pit. Manures from swine fed high fiber diets generally have increased methane production rates and organic N content [29, 30], both of which were significantly associated with manure foaming (Table 1). These results suggest the feed ingredients' digestibility impacts the formation of different manure surface textures. Foaming was associated with less digestible diets, and crust-

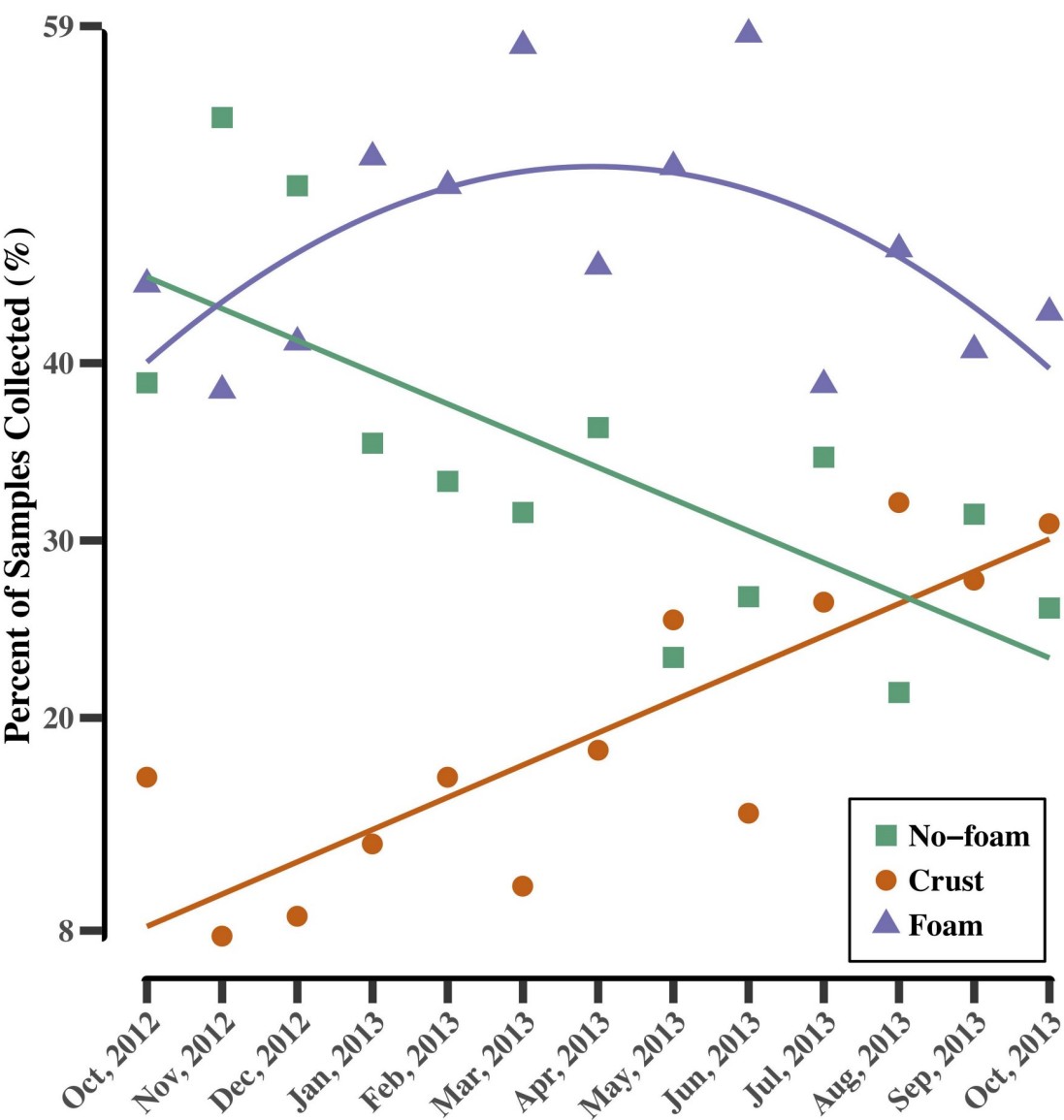

**Fig 1. The proportions of manure samples with different surface textures collected over 13 months, where "no-foam" represents non-foaming manure, "crust" represents crust-forming manure, and "foam" represents foaming manure.** The lines are the fitted trend lines showing the changes in percent of different type of samples collected over the study.

forming manure was associated with high crude protein, crude fiber, and acid detergent fiber (ADF) (Table 1).

## Significant correlations between different carbon compounds observed in deep pit manure

The formation of no-foam, crust, or foam manure-surfaces is likely due to various physical-chemical interactions. Thus, we identified the most significant correlations among manure physical-chemical properties (S4 Fig). We found that manure physical properties, especially a higher surface tension and lower foaming capacity, had the strongest correlations with non-foaming manures. In contrast, strong correlations among chemical properties were observed

**Table 1. Diet and manure characteristics in manures with different surface textures.**

| Significant Contributors | Number of Samples | | | Observed Trends | $BF_{10}$ | P-value |
|---|---|---|---|---|---|---|
| | No-foam | Crust | Foam | | | |
| Soybean Meal[a] | 157 | 95 | 228 | No-foam > Crust > Foam | 24.25 | 0.0396 |
| Crude Protein[b] | 157 | 95 | 228 | Crust > Foam > No-foam | 23.94 | 0.0401 |
| ADF[b] | 157 | 95 | 228 | Crust > Foam > No-foam | 656.57 | 0.0015 |
| Crude Fiber[b] | 157 | 95 | 228 | Crust > Foam > No-foam | 2358.38 | 0.0004 |
| Manure Temperature | 149 | 96 | 223 | Crust > Foam > No-foam | 35922.64 | < 0.0001 |
| Manure Depth | 162 | 98 | 228 | Crust > Foam > No-foam | 5.42E+08 | < 0.0001 |
| DDGS[a] | 157 | 95 | 228 | Foam > Crust > No-foam | 40.23 | 0.0243 |
| NDF[b] | 157 | 95 | 228 | Foam > Crust > No-foam | 1920.51 | 0.0005 |
| $CH_4$ Production Rate (slurry) | 143 | 77 | 176 | Foam > Crust > No-foam | 1.32E+09 | < 0.0001 |
| Organic N | 141 | 77 | 172 | Foam > No-foam > Crust | 23.27 | 0.0412 |

a. Major diet ingedients and supplements.

b. Diet components as formulated.

c. P value calculated as $p(H_0|D)$.

Bayes factors are in column $BF_{10}$. Column P-value shows the posterior likelihood of observed trends not occurring.

in foaming manures. Specifically the correlations among free long chain fatty acids (LCFA), short-chain fatty acids (SCFA), carbon content, and pH (Fig 2). Crust manures were correlated with physical and chemical measurements, such as surface tension and high manganese and potassium content. These observations suggest that the manure surface texture change from no-foam to foam is associated with a shift from the dominance of correlations among physical properties to correlations among chemical properties, with crusts as an intermediate.

Chemical interactions have previously been associated with foaming. Yan et al (2015) found that the concentration of LCFA were the greatest in the manure's foam layer, with lower LCFA concentration observed in the liquid portion of the foaming manure compared to non-foaming manure [31]. In this study, we only evaluated the concentration of LCFA in the liquid portion of the manure. We found that the LCFA from the swine manure were predominately C16 and C18 compounds. Consistent with the previous findings, the average concentration of LCFA in the foaming manure liquid was lower than the non-foam manure liquid, although the difference was not significant. However, the concentration of LCFA was significantly enriched in crust-forming manure, with 15 and 22 times more observed than no-foam and foaming manure, respectively. This is surprising as animals associated with the foaming manure were fed significantly more DDGS, which contains more lipids than other diet ingre-dients, such as SBM (Table 1) [6]. Dietary lipids are main sources of LCFA and it is natural to expect the amount of LCFA measured corresponds to the amount of lipids received. However, our observation that the concentration of LCFA in manure did not correlate with the amount of DDGS in diet suggests that the accumulation of LCFA in manure alone is insufficient to explain the manure foaming.

Acetic acid comprised more than 50% of the total SCFA measured from the pit manure samples, with propionic and butyric acid being the second and third in abundance. The pro-portion of each SCFA out of the total amount of SCFA detected did not differ significantly among manure types. However, in foaming manure, a significant strong positive correlation between LCFA and SCFA was observed, which was absent in no-foam and crust manure (Fig 2). This observation highlighted the potential for the chemical conversion of LCFA to SCFA as an essential step in forming foaming manure. Further, it would be consistent with the

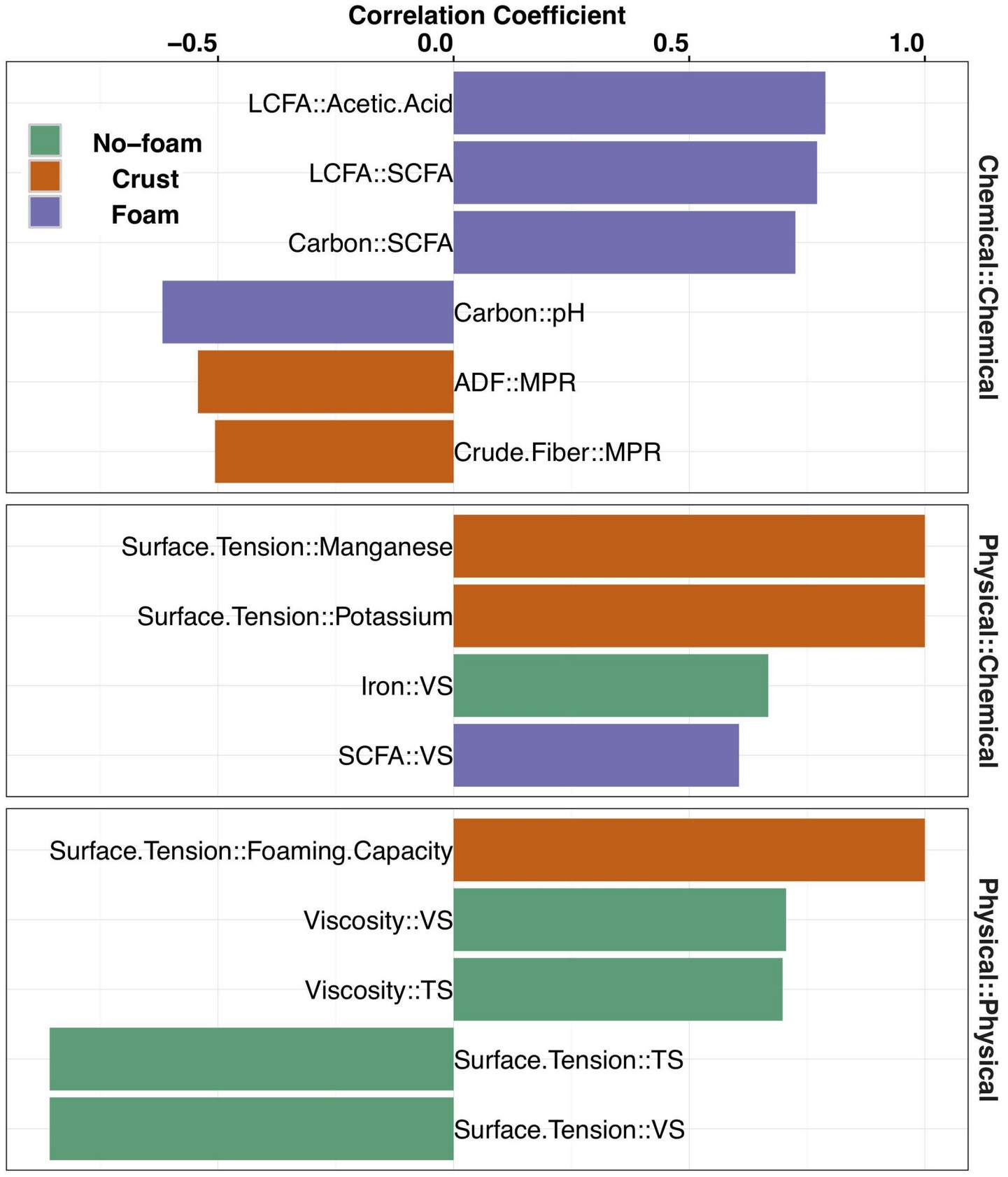

**Fig 2. The five strongest manure characteristic associations in foam (purple), crust (orange), and no-foam (green) manure samples.** Bars extending towards left represent negative correlations and bars extending towards right represent positive correlations.

enrichment of LCFA only in crust manure, as it may be converted efficiently to SCFA in foaming manure. Generally, SCFA, including acetic acid, and LCFA are microbial metabolites produced by gastrointestinal microorganisms and are important nutrients consumed by host animals and gastrointestinal microorganisms [32–34]. As fecal matter is excreted into manure pits, these fatty acids likely continue to support microbial metabolism in manure pits. While various microorganisms directly assimilate SCFA, LCFA generally cannot be readily utilized by most microbes until it is degraded to SCFA [35–37]. The significant correlation between LCFA and SCFA in foaming manure suggests that the foaming manure-associated microorganisms are efficient at converting LCFA to SCFA and supporting the growth of methanogens. The study limitation is that only the endpoint accumulations of LCFA and SCFA were measured, which do not reflect the rate of depletion and production. However, our results indicate a clear hypothesis that increased conversion of LCFA to SCFA would result in foaming in manure pits. This observation is also consistent with previous results indicating that the degradation of LCFA to SCFA is an important fermentation step in methane production from lipids [38, 39].

There were significantly more SCFA (p = 0.0046) and acetic acid (p = 0.0014) in the no-foam manure than foaming manure. On average, 12.4 mg/g and 7.3 mg/g of SCFA was detected in the no-foam and foaming manure, respectively, while 7.5 mg/g and 4 mg/g of acetic acid was detected in the no-foam and foaming manure, respectively. SCFA is not the thermodynamically preferred microbial fermentation end-products and SCFA accumulation is known to inhibit methanogenesis in anaerobic fermentation [40]. Consequently, the MPR in non-foaming manure was significantly lower than those of foaming manure (Table 1). Although a large quantity of SCFA could reduce manure pH, which may impact methanogenesis as well [41], the pit manure pH averaged at 8.2 and did not differ significantly between manure types. Thus, SCFA likely had little influence on the manure pH and the manure pH did not contribute to the observed MPR differences among manure types.

Under anaerobic conditions, the breakdown of LCFA is carried out via acetogenesis [35, 42]. This process is endogenic and does not occur spontaneously under standard conditions. However, by coupling methanogenesis, an excess amount of SCFA can be removed and result in an overall exogenic reaction [42]. Therefore, methanogenesis is an important step in efficient anaerobic LCFA degradation. Together with our observations, manure foaming is associated with efficient anaerobic LCFA degradation with two interlinked components: acetogens that break down LCFA to SCFA and methanogens that remove SCFA and turn the overall reaction spontaneous. In addition, recent studies found that anaerobic fatty acid chain elongation could effectively conserve energy in the absence of methanogens [43, 44]. Thus, this could be an alternative SCFA processing pathway in no-foam manure if methanogenesis was indeed inhibited and should be evaluated further in future studies.

## Foam, no-foam, and crust samples contain distinct bacterial and methanogenic communities

Consistent with the observation of varying metabolites in the varying manure textures, the microbial community composition in foaming and non-foaming manures significantly differed from each other (Fig 3). Variations among microbial communities associated with the different manure types were identified by sequencing the 16S rRNA gene and methyl coenzyme M reductase (*mcrA*) gene, conserved phylogenetic markers in bacteria and methanogens,

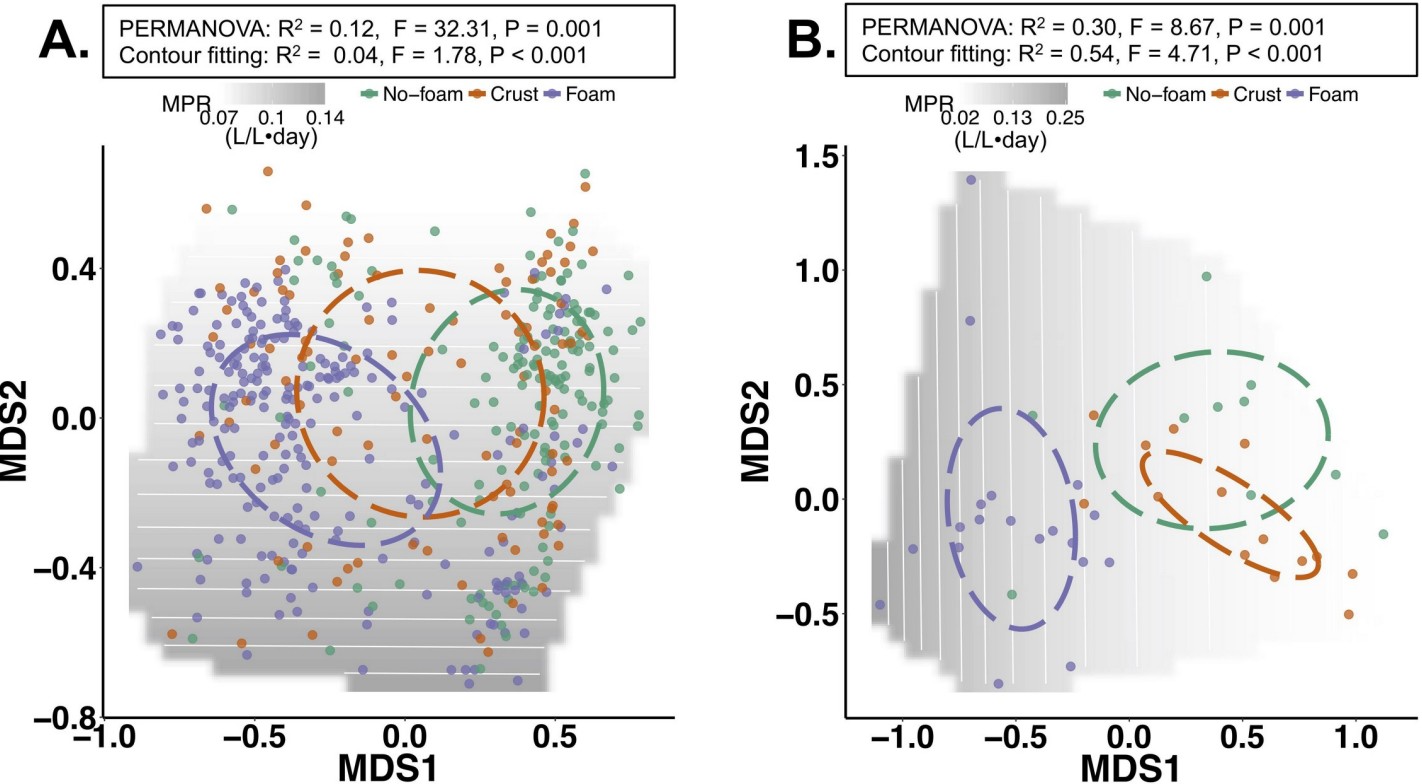

**Fig 3.** Non-metric multidimensional scaling (NMDS) analysis of bacterial communities (panel A) and methanogen communities (panel B) by the Bray-Curtis distances calculated using the relative abundance of microbial operational taxonomic units (OTU). The ellipses represent 95% confidence level around the centroids of manure samples with different surface textures. The microbial community variations among manure samples with different surface textures were assessed using Permutational Multivariate Analysis of Variance Using Distance Matrices (PERMANOVA). The methane production rates (MPR) were modeled to overlay the observed community differences (Contour fitting). The grey background shows the fitted MPR based on the measured MPR, with darker grey represents higher MPR.

respectively. Bray-Curtis dissimilarity revealed that the farm where the manure sample originated contributed to the largest variation in microbial community structures ($R^2 = 0.48$ for bacteria, $R^2 = 0.83$ for methanogens) (S3 Table). This is consistent with the findings of previous studies where individual storage tank explained the manure microbiome the best, better than diet, suggesting that the unique microbiome of each manure storage tank is contributing to the manure condition [5, 6]. Given the high variability among farms, we treated individual farms as experimental blocks and found that bacterial communities differed significantly, with the greatest variation ($P = 0.001$, $R^2 = 0.12$) among no-foam and foaming manures (Fig 3A). Similar significance patterns were also observed in methanogenic communities ($P = 0.001$, $R^2 = 0.30$) (Fig 3B).

Next, we examined whether the microbial communities of a specific manure type had strong correlations with MPR. We compared MPR measurements to the distribution of observed taxa containing the *mcrA* gene in manure samples. The correlation between MPR and microbial community distribution revealed a strong association between increased MPR and manure pit methanogenic communities ($R^2 = 0.54$, $p < 0.001$, Fig 3B). The methanogen communities of foaming manure differed from no-foam and crust-forming manure significantly ($R^2 = 0.3$, $p = 0.001$) and was associated with high MPR, suggesting foaming manure may contain unique methanogens. Notably, although bacterial communities of non-foaming, crust-forming, and foaming manure also differed significantly ($R^2 = 0.12$, $p = 0.001$), the bacterial community differences did not correlate well with MPR ($R^2 = 0.04$) (Fig 3A). Overall,

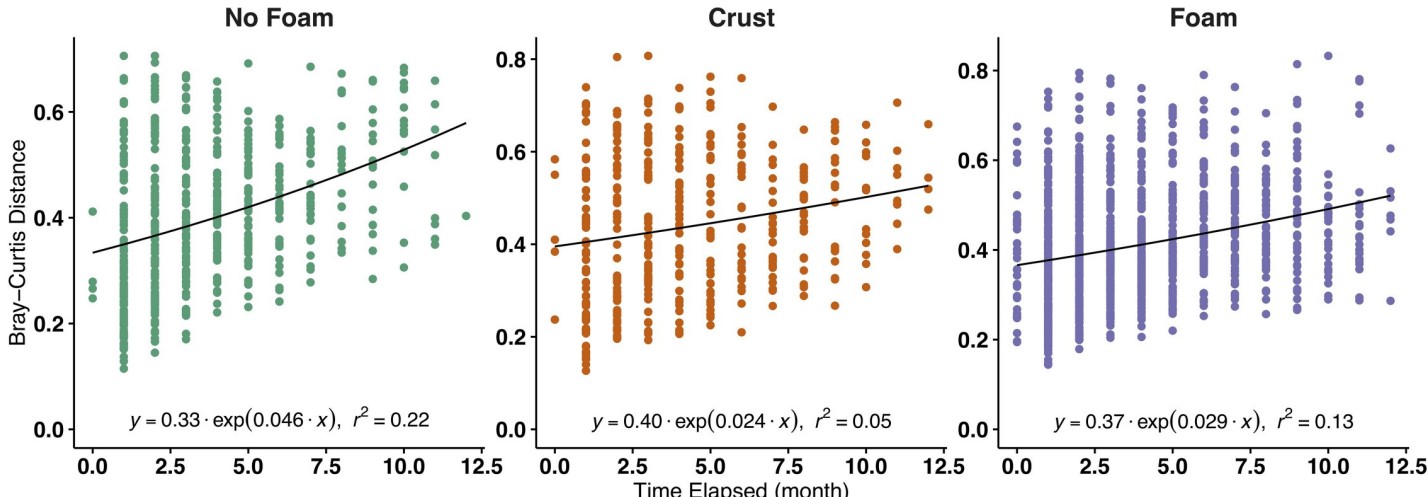

**Fig 4.** The manure-associated community dissimilarity as a function of time (y = a•exp(b•x)), where a smaller slope (b) suggests a smaller dissimilarity over time. Distribution of slopes was estimated by bootstrapping each group of samples 999 times. The slope estimated for non-foaming samples was significantly greater than those estimated for crust-forming and foaming samples (overlap coefficient = 0.0467).

these results suggest that MPR in swine manure storage pits is significantly related to methanogen community compositions. The bacterial communities are not directly related to methane production but may indirectly contribute to the stability or persistence of the foam.

Overall, the bacterial communities in foaming manure and crust-forming manure were significantly more stable than those in non-foaming manure (Fig 4). This result is consistent with our observation that pits, once foaming, persist with foam continuously (S1 Fig). Andersen et al. (2018) previously found that the bacterial communities of no-foam manure were more prone to changes upon an antibiotic's addition (i.e., ionophore). In contrast, the foaming manure communities were much more resistant to this perturbation [45]. This study further corroborates their findings and suggests foaming-manure microbial communities are less likely to shift with disturbances.

To minimize the community variations unique to individual farms, microbial OTUs present in all samples associated with the same surface texture were selected to represent core no-foam, crust, and foam bacteria communities (e.g., "core" no-foam, crust, and foam communities). The number of shared and unique bacterial and archaeal taxa among the three manure textures that were significantly different in observed abundances were identified (Fig 5A, 5B). Shared between foaming and non-foaming manures were taxa associated with the phyla Bacteroidetes, Firmicutes, Proteobacteria, and Spirochaetes. These phyla were broadly present in manures, but their proportional abundances at the genus level differed significantly ($BF_{10} >= 100$) among no-foam, crust, and foam manures (Fig 5C), suggesting that different bacterial assemblages are associated with the varying manures. For example, broadly distributed in all manure samples were genes related to the archaeal phylum Euryarchaeota and specific unclassified archaea, but their specific abundances varied in different manure types. At the species level, the Euryacheaota *Methanosphaera stadtmanae* was significantly more abundant in no-foam manure ($BF_{10} >= 20$), and the unclassified archaea were the most abundant in foam manure ($BF_{10} >= 20$) (Fig 5D).

Generally, we observed that deep-pit manures contained high acetic acid (average 4.4 mg/g). However, no sequences related to acetic acid-degrading methanogens, specifically Methanosarcinaceae- or Methanosaetaceae-affiliated *mcrA* genes, were detected. Members of these

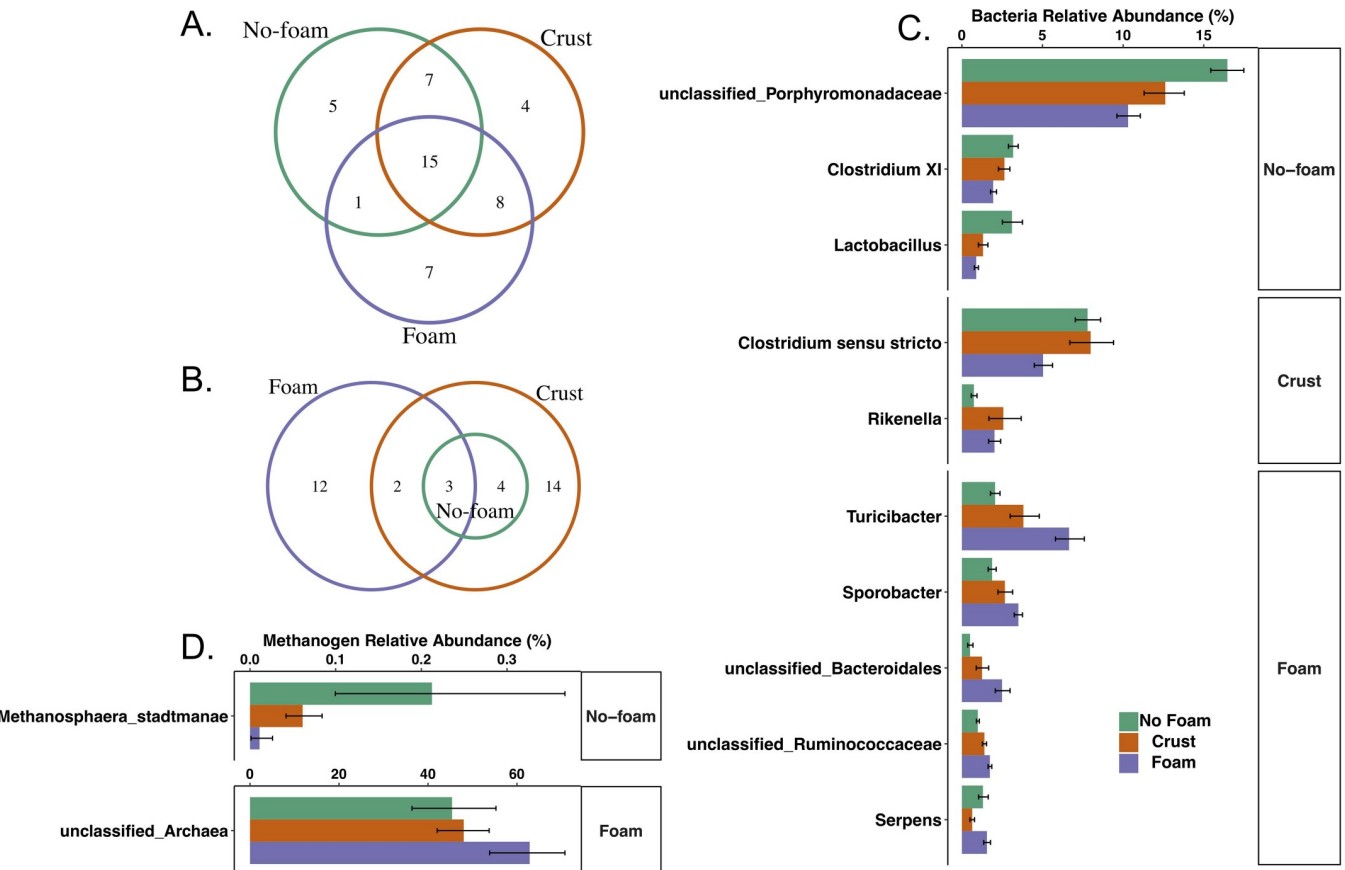

**Fig 5.** The distribution of shared and unique A) bacterial and B) methanogenic core of no-foam, crust, and foam manure samples. The numbers represent the operational taxonomic unit (OTU) counts. Panel C shows the ten most relatively abundant bacterial groups at the genus level where relative abundances were significantly different among no-foam, crust, and foam manures. Methanogen groups that differed significantly in relative abundance among different types of manure are shown in panel D. Individual bar represents the average of a microorganism relative abundance in the specified manure type, while individual error bar represents the 95% confidence interval calculated using bootstrapping method. Within panel C and D, the labels on the right side indicate the manure type in which the bacterial or methanogen groups were the most relatively abundant in.

two families are the only known methanogens that can directly produce methane by cleaving acetic acid (acetoclastic methanogens) and are usually enriched in high acetate environments [46, 47]. Instead, hydrogenotrophic methanogens, such as Methanobacteria and Methanomicrobia, were abundant in the manure storage pits. Studies have found that ammonium can inhibit the growth of acetoclastic methanogens, and in an environment with a large amount of acetate but a lack of acetoclastic methanogens, acetate oxidation in conjunction with hydrogenotrophic methanogens may play an important role in anaerobic methane production [47, 48]. Therefore, methane production in swine manure store pit is likely solely carried out by hydrogenotrophic methanogens.

To better evaluate these taxa interactions and the metabolites associated with manures, relationships between relative taxa abundances and results from manure chemical analysis were explored. In no-foam manure samples, the SCFA concentration was positively correlated with a Bacilli (Lactobacillus) OTU and negatively correlated with Clostridia members (Fig 6). This result is in line with previous observations that lactic acid-producing bacteria may produce SCFA and subsequently inhibit the growth of Clostridia members [49, 50]. The manure SCFA content was also negatively correlated with a *Turicibacter sp.* (Erysipelotrichia member) in no-

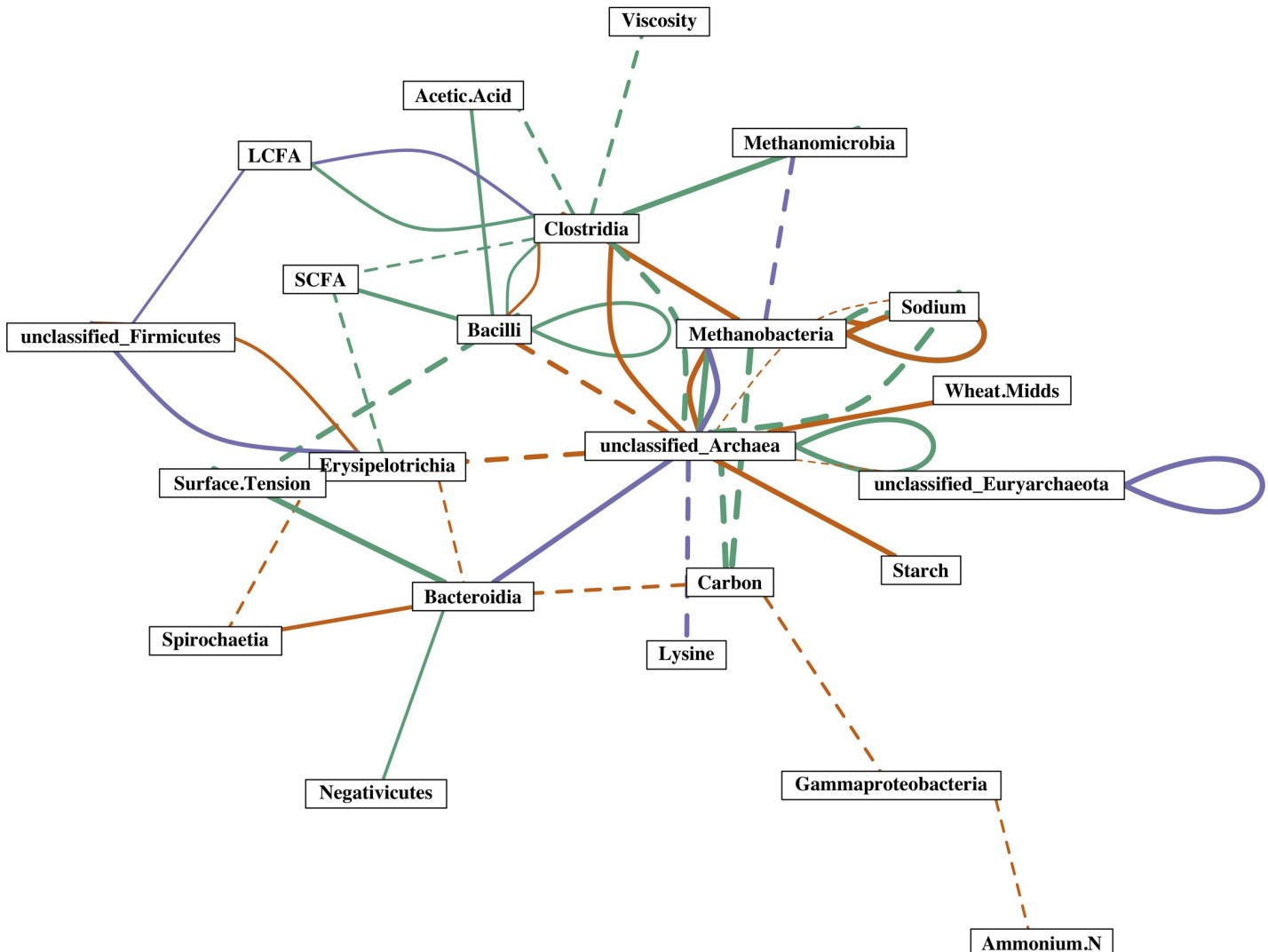

**Fig 6. The core significant and strong correlations between bacteria and methanogens, bacteria/methanogens and dietary inputs, and bacteria/methanogens and manure characteristics.** Individual rectangle labels represent dietary input, manure measurements, or bacterial and methanogenic groups at the class level. The solid lines represent positive correlations and the dashed lines represent negative correlations. A thicker line indicates a stronger correlation. Correlations observed in no-foam, crust, and foam manure were shown in green, orange, and purple, respectively. Looped correlations indicate that relationships were observed among members of the same microbial groups.

foam manure. Little is known about *Turicibacter*, except that they have been previously identified as important animal microbiota members[51]. In contrast to no-foam manures, no significant correlations between individual bacterial taxa and SCFA content were identified in foam manure (Fig 6).

Different Clostridia members were positively correlated with the concentration of LCFA in both non-foaming and foaming manure, *Sporobacter sp.* in foaming manure, and *Sporobacterium sp.* in non-foaming manure (Fig 6). Unique to foaming manure, a single unclassified Firmicutes OTU was also positively correlated with the concentration of LCFA. Although *Sporobacter* and *Sporobacterium* members were previously reported in lipid abundant methane-producing anaerobic systems, they are not known to be LCFA degraders [52–54]. Therefore, we suspect that the unclassified Firmicutes played a role in converting LCFA to SCFA in

the foaming manure. The lack of correlation between LCFA and bacteria in the crust-forming manure is consistent with the observation of LCFA accumulation, suggesting little degradation of LCFA occurred in the crust-forming manure.

Intriguingly, Clostridia were positively correlated with Methanomicrobia and Methanobacteria in both no-foam and crust-forming manure, respectively, while Bacteroidia were positively correlated with unclassified Archaea in foaming manure. While members of both Clostridia and Bacteroidia are known hydrogen producers that support methanogenesis, a high abundance of Bacteroidia has previously been found in high methane-producing anaerobic bioreactors [55–57]. The synergistic interactions between bacteria and methanogens could explain the uniquely high MPR observed in the foaming manure, despite hydrogenotrophic methanogenesis being the main methane-producing pathway in all manure storage pits in the study.

## Conclusions

In summary, our key findings based on characterizing the foaming conditions of manure pits in 46 Iowa farms are: (1) foaming is associated with increased levels of indigestible fiber; (2) manure LCFA concentration is strongly correlated with the manure SCFA concentration in foaming manure only, suggesting direct conversion from LCFA to SCFA was an essential step in manure foaming; (3) different synergistic interactions between bacteria and methanogens correlated with different methane production rates; (4) specific taxa are associated with foaming, non-foaming, and crust manures; (5) a lower MPR in non-foaming pits is likely due to both the accumulation of SCFA and a less efficient combination of bacteria/methanogen group; (6) foaming microbial communities are relatively more stable compared to non-foaming microbial communities, suggesting the development of a mutualistic microbial relationship in the foaming manure.

Based on the observation of specific taxa in different manure textures, we hypothesize that there are differences in the efficiency of manure organic matter anaerobic fermentation involving the degradation of LCFA (Fig 7). Specifically, we observed that the microbial community of foaming manure can completely degrade manure organic matter to methane via LCFA and SCFA conversion, as evidenced by no significant accumulations of intermediate metabolites. In contrast, in crust-forming manure, manure organic matter fermentation is stalled during LCFA degradation, and we observed a significant LCFA accumulation. No significant correlation was observed between the concentrations of LCFA and SCFA in no-foam manure; however, the significant accumulation of SCFA may be due to the production of an excess amount of SCFA from manure organic matter or the inefficient utilization of SCFA in no-foam manure. The limitation of this study is the lack of fatty acid rate of production and consumption, which should be investigated in future studies. Despite the limitation, our hypothesis expands upon our key findings that foaming is most likely directly related to specific unclassified methanogens and their relationship with specific hydrogen producing bacteria, which are supported by metabolites in manures and their interactions with feed fiber. For management of foaming manures, future research would benefit from understanding the stability of the foam. We suggest that there are key taxa that are involved in the conversion (Firmicutes, foaming) or lack of conversion (Lactobacillus, non-foaming) of LCFA to SCFA. We highlight these taxa as potential membership that can be used as resources for foaming manure management. For example, the impacts of the addition of Lactobacillus for mitigating the transition of non-foaming to foaming pits would be a key area for future research. The observed stability of the foaming manure and its microbial communities present risks to managing foam and its methanogen production. For long term manure storage, these risks are further heightened as

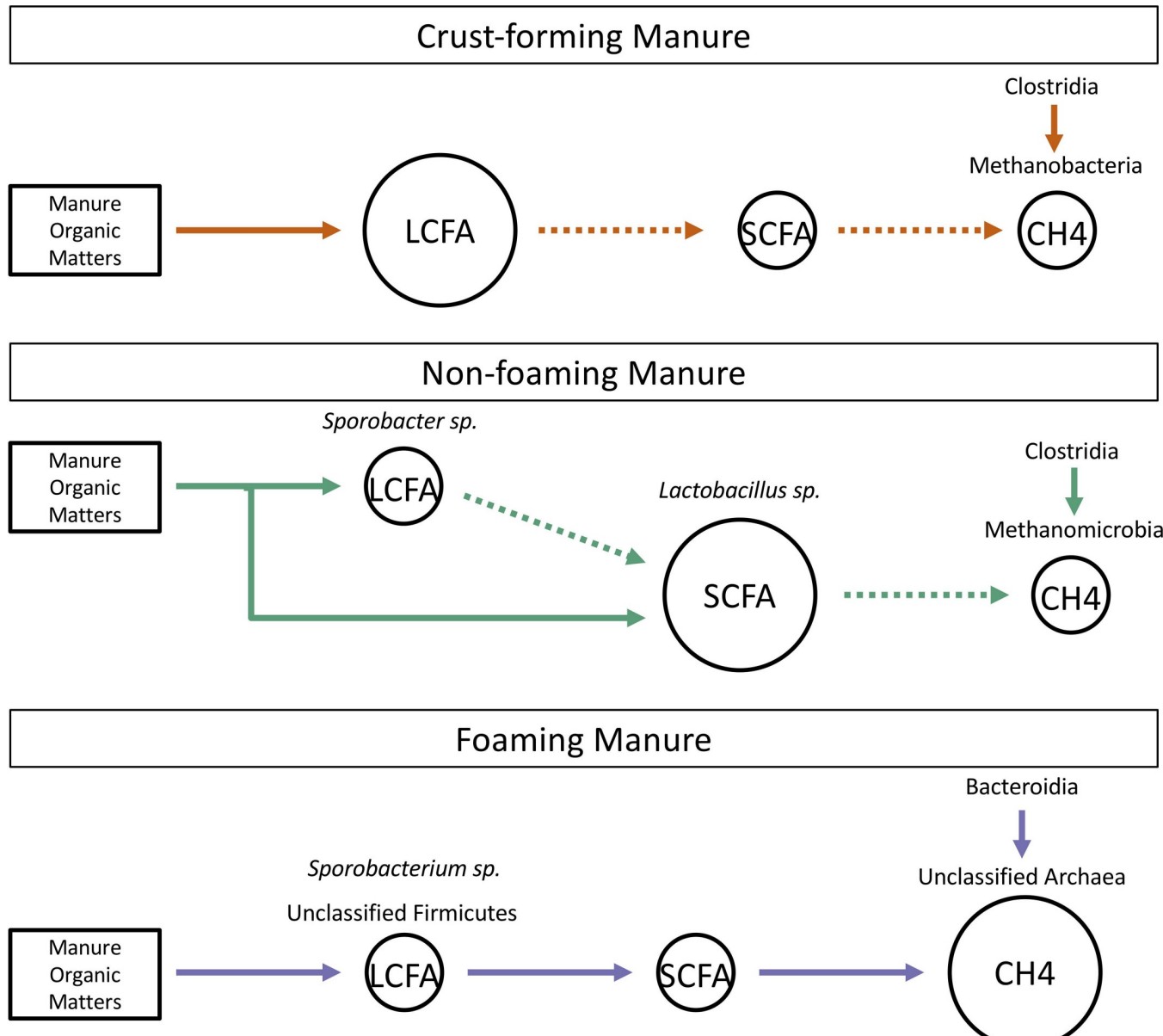

**Fig 7. The predicted manure fermentation processes in crust-forming, non-foaming, and foaming manure.** The solid and dashed lines represent efficient and inefficient processes, respectively. The circles represent the fermentation by-product and the large circle indicates the accumulation of the by-product. The microorganisms that strongly and positively correlated with the by-product are listed above the circles.

trends indicate that DDGS will have increased use in feed [58]. In this regard, the identification of alternative resources, such as microbial additions or treatments, would be of value for providing safe and sustainable manure management.

## Supporting information

**S1 Fig. The manure samples and their surface textures collected from 46 farms from October 2012 to October 2013.**
(TIFF)

**S2 Fig. Manure samples with different surface textures, from left to right: Non-foaming, non-foaming, foaming, foaming, crust-forming, and foaming manure.**
(TIFF)

**S3 Fig. The vertical profile of manure in the storage facilities.** Layer A describes manure surface texture. Manure characterizations and bacterial community analyses were performed on samples from layer B. Samples from layer C were used to measure methane production rates (MPR) and for methanogen community analyses.
(TIFF)

**S4 Fig. The significant correlations among dietary information and manure characteristics.** The major dietary inputs were separated into ingredients and supplements (red bars) and nutrients (blue bars). Green bars represent manure characteristics. Within each grid, the larger the square box, the stronger the correlation. The square box color in each grid shows the type of manure the correlation was observed in. A grid with multiple boxes suggests it is a common correlation found in manure with different surface textures. A grid with a single box suggests it is a unique correlation found in that particular type of manure as the color indicated.
(TIFF)

**S1 Table. General information of farms where manure samples were collected from.**
(PDF)

**S2 Table. Final number of samples used in the study after removing samples with missing manure characteristics and dietary information and samples with low sequencing coverage.** The final average bacterial sequencing depth and coverage were reported in columns 3 and 4. The final average methanogen sequencing depth and coverage were reported in columns 6 and 7.
(PDF)

**S3 Table. Permutational multivariate analysis of variance of microbial communities with different experimental factors.** Bray-Curtis dissimilarities were calculated using operational taxonomic unit (OTU) relative abundance.
(PDF)

**S1 File. Additional supporting analysis methods.**
(DOCX)

## Acknowledgments

We would also like to thank the staff at Roy J. Carver Biotechnology Center of University of Illinois at Urbana Champaign for generating the sequences, and Caleb Polson and Mark Van Weelden for additional sample analyses.

## Author Contributions

**Conceptualization:** Fan Yang, Daniel S. Andersen, Steven Trabue, Angela D. Kent, Richard S. Gates, Adina S. Howe.

**Data curation:** Fan Yang, Daniel S. Andersen, Steven Trabue, Angela D. Kent, Laura M. Pepple, Adina S. Howe.

**Formal analysis:** Fan Yang, Daniel S. Andersen, Steven Trabue.

**Funding acquisition:** Daniel S. Andersen, Steven Trabue, Angela D. Kent, Richard S. Gates, Adina S. Howe.

**Investigation:** Fan Yang, Daniel S. Andersen, Steven Trabue, Angela D. Kent, Laura M. Pepple, Richard S. Gates, Adina S. Howe.

**Methodology:** Fan Yang, Daniel S. Andersen, Steven Trabue, Angela D. Kent, Laura M. Pepple, Richard S. Gates, Adina S. Howe.

**Project administration:** Daniel S. Andersen, Steven Trabue, Angela D. Kent, Richard S. Gates, Adina S. Howe.

**Resources:** Fan Yang, Daniel S. Andersen, Steven Trabue, Angela D. Kent, Laura M. Pepple, Richard S. Gates, Adina S. Howe.

**Software:** Fan Yang, Adina S. Howe.

**Supervision:** Daniel S. Andersen, Angela D. Kent, Richard S. Gates, Adina S. Howe.

**Validation:** Fan Yang, Daniel S. Andersen, Steven Trabue, Angela D. Kent, Laura M. Pepple, Richard S. Gates, Adina S. Howe.

**Visualization:** Fan Yang.

**Writing – original draft:** Fan Yang, Daniel S. Andersen, Steven Trabue, Adina S. Howe.

**Writing – review & editing:** Fan Yang, Daniel S. Andersen, Steven Trabue, Angela D. Kent, Laura M. Pepple, Richard S. Gates, Adina S. Howe.

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
