## [Decision Letter · Decision Letter 0]

9 Apr 2021

PONE-D-21-06556

Microbial assemblages and methanogenesis pathways impact on methane production and foaming in manure deep-pit storages

PLOS ONE

Dear Dr. Yang,

Thank you for submitting your manuscript to PLOS ONE. After careful consideration, we feel that it has merit but does not fully meet PLOS ONE’s publication criteria as it currently stands. Therefore, we invite you to submit a revised version of the manuscript that addresses the points raised during the review process.

We look forward to receiving your revised manuscript.

Kind regards,

Alex V Chaves, PhD

Academic Editor

PLOS ONE

We note that one or more of the authors are employed by a commercial company: Puck Custom Enterprises.

2.1. Please provide an amended Funding Statement declaring this commercial affiliation, as well as a statement regarding the Role of Funders in your study. If the funding organization did not play a role in the study design, data collection and analysis, decision to publish, or preparation of the manuscript and only provided financial support in the form of authors' salaries and/or research materials, please review your statements relating to the author contributions, and ensure you have specifically and accurately indicated the role(s) that these authors had in your study. You can update author roles in the Author Contributions section of the online submission form.

2.2. Please also provide an updated Competing Interests Statement declaring this commercial affiliation along with any other relevant declarations relating to employment, consultancy, patents, products in development, or marketed products, etc.  

Reviewers' comments:

Reviewer's Responses to Questions

**Comments to the Author**

1. Is the manuscript technically sound, and do the data support the conclusions?

Reviewer #1: Partly

Reviewer #2: Yes

Reviewer #3: Yes

2. Has the statistical analysis been performed appropriately and rigorously? 

Reviewer #1: Yes

Reviewer #2: Yes

Reviewer #3: Yes

3. Have the authors made all data underlying the findings in their manuscript fully available?

Reviewer #1: No

Reviewer #2: Yes

Reviewer #3: Yes

4. Is the manuscript presented in an intelligible fashion and written in standard English?

Reviewer #1: Yes

Reviewer #2: Yes

Reviewer #3: Yes

5. Review Comments to the Author

Reviewer #1: The authors compared physical and chemical properties of different types of swine manure pit samples (no-foam, foam, and crust), and made associations of these with the bacterial and methanogen community structure, and methane production rate. The work is interesting and generally well-performed, and it does shed light on some of the differences in these sample types. Despite these advances, the reviewer feels that the authors could have extended their observations by delving a bit more into the characteristics of the samples, particularly with regard to the SCFA and LCFA (see Specific comments below), as both types of compounds are major substrates or intermediates and their metabolic fates have a strong influence on methane production and, presumably, microbial community structure.

A second issue regards the pH of the samples, and how it might relate to SCFA concentration and to methane production. It appears from L66 and Supplemental Figure S3 that pH was measured, and there was a slight correlation of pH with some of the other measured characteristics (particularly nutrients), especially in the no-foam manures, but little correlation of pH with SCFA. This should perhaps be pointed out somewhere in the text. It would also be useful to indicate the range of pH values encountered, and whether they differed significantly among the manure types. As the aceticlastic methanogens tend to be more sensitive to acidic conditions than are the hydrogenotrophic methanogens, could pH explain the general absence of the former group in these samples?

Specific comments:

L50: The authors bring up the tendency of foaming to cause a potential for explosion, but in the Results we learn that the foaming samples showed lower rates of methane production than did the other samples. Presumably the foaming causes methane accumulation even if it is produced in lesser amounts. This should be mentioned in the Discussion.

L64-67: The methods description is inadequate. Reference 4 does not provide analytical methods, and no methods are described for LCFA and SCFA analysis.

L115, L126: How were these sequence numbers (10k for bacteria, 4k for methanogens) selected, as opposed to normalizing the number of reads selected to number of reads in the sample with the fewest reads?

L158-161: It would be very useful to be more specific regarding these sample types, perhaps by including photographs (even if included in Supplementary data).

L181-183: Another important difference between SBM and DDGS is the amount and type of LCFAs, which should have some bearing in discussion of the importance of LCFA and its metabolism in the three sample types.

L209-211: This seems like a superficial distinction, as the “physical interactions” must have some sort of underlying chemical basis (e.g., surface tension is probably governed by the relative contributions and hydrophobic/hydrophilic behavior of proteins, lipids and polysaccharides).

L228-254: In this section the authors discuss SCFA and acetic acid in the samples without providing any data on the amounts and distributions (i.e., different chain lengths) of these acids. Such information would be very useful in interpreting potential bottlenecks in LCFA degradation to SCFA, and SCFA catabolism to methane.

L256: This is an oversimplification. Perhaps the authors should add “under standard conditions”. The process of LCFA degradation to SCFA typically operates near equilibrium, so that it is quite sensitive to the relative concentrations of substrate and product. But it does not typically require input of energy, and the authors seem to suggest this in the following sentences, in which they state that methanogenesis provides thermodynamic displacement via H2 consumption, allowing SCFA catabolism to occur.

L320-321: It is not clear to the reviewer how the individual slope values for the three types of manures can be compared to yield statistically significant differences, as there is only a single slope value for each sample type..

L349-360: These are interesting observations, but they raise the question of where the acetic acid goes. Working back to comments to L228-254 above, is it possible that some members of the bacterial community might be engaged in chain elongation reactions, in which acetate is converted to longer SCFA such as butyrate and caproate? Chain elongation has been widely observed in waste pits and anaerobic reactors in which aceticlastic methanogenesis is limited (for example, see Zhu et al., DOI: 10.1038/srep14360, or Angenent et al., DOI: 10.1021/acs.est.5b04847).

L398-401: What is meant here by the vague statement regarding “the different potential involvement of bacteria or methanogens”?

L421-424: The reviewer is not convinced by this statement. Why can’t the accumulation of SCFA be due to impaired catabolism of SCFA, rather than invoking a greater production of SCFA from other feed components?

Table S2: Please define (in a footnote) the term “integrator”.

Minor edits:

L45-46: Delete “However,” and use rest of sentence as the first sentence of the next paragraph.

L50: Delete “conditions of”.

L185: Change “low” to “poorly”.

L229: Here and elsewhere in the manuscript, change “SCFA/acetic acid” to just “SCFA”, because (as pointed out in L233-234) acetic acid is one of the SCFA.

L277: Change “between” to “among”.

L329: Change “genera” to “genus”.

L347: Change “that the bacterial or methanogen groups were the most abundant in” to “in which the bacterial or methanogen groups were the most abundant”.

Reviewer #2: Manuscript entitled "Microbial assemblages and methanogenesis pathways impact on methane production and foaming in manure deep-pit storages" can be considered as high importance. This manuscript provides relatively new and unique dataset to tackle the human, animal and environmental risks associated with swine industry. The study provided a relatively large dataset to support the finding. Manuscript is well written and deserve publications.

Few things, however, authors are suggested to consider: 1) provide the details of farm (herd size; pit size; and existing manure management in farms where samples were collected); 2) Figure 1 requires further descriptions in Figure captions (currently not clear); 3) method section requires further clarifications improvement in terms of sampling protocol, and the strategies implemented in sample collection; 4) before ending results and discussion section, authors are suggested to provide major findings quantitatively, and limitations of the study; and 5) discussion on the composition of gas formation, ammonia, and microbial community identified in Figure 5,6.7. Any pathogens, in addition to methanogens were identified?

Further, in statistics table, authors are suggested to provide the sample numbers used for the analysis. In terms of feed characterization, not sure if the feed of individual farms has been characterized or the feed from mills which supplies the feed has been characterized. Each pit will have certain holding capacity, and the retention time will be having impacts of microbial community. Therefore, the conditions of pits (mixing, retention time, temperature, flushing frequency, and the frequencies of disloading of those pits are important consideration while describing the shift in microbial community under anaerobic conditions.

Reviewer #3: This manuscripts describes the bacterial and methanogenic communities of manure collected from 46 swine farms at several different time points. The authors visually classified the manure surface as either non-foaming, crust-forming or foaming at each sampling time.

Certain chemical properties of the manure were also assessed. The authors report that the swine diet influences the manure surface texture and that certain chemical and physical properties were correlated with the manure surface texture as well. The non-foaming and foaming manure bacterial and methanogenic microbial communities were most dissimilar from each other.

Major comments

The materials and methods for the manure sampling and chemical analysis appear to be missing. The superscript “4” is included at the end of the sentence on ln 65 but it is unclear if this is referring to REF 4 or something else.

Minor comments

In Table S2 should the P-values for Farm and Feed Mill for bacteria also be 0.001?

Can the x-axis in Fig. 5 C and D be converted to percent relative abundance rather than the current scale which is not very intuitive.

Ln 24: “an increased”

Ln 26-27: “…rRNA and archaeal mcrA genes…”

Ln 29: “and that the bacterial…”

Ln 31: “previously” should be removed since they are still uncharacterized.

Ln 33, 362: “relative abundance”

Ln 50: I would remove “conditions of”

Ln 81: “mcrA gene was”

Ln 111-113: What database was used for 16S rRNA gene classification?

Ln 121: “chimeras”

Ln 180: What is meant by “more significant” here? A greater concentration of DDGS? Also, what does the superscript “19” refer to here?

Ln 182: “NDF” should be first defined here rather than on ln 185.

Ln 205, 407: “were correlated”

Ln 228: The use of “SCFA/acetic acid” here and elsewhere is unclear to the review. Was acetic acid the only SCFA measured?

Ln 275: Remove “occurrence of”

Ln 338: “of shared and unique A) bacterial and B) methanogenic OTUs of no-foam…”

Ln 340: “most relatively abundant”

Ln 342: “that differed”; “are shown”

Ln 346: “relatively abundant”

Ln 361: “taxa”

Ln 364: “was positively correlated”; “Bacilli (Lactobacillus) OTU and…”

Ln 366: I would remove “members of”

Ln 367-368: “…content was also negatively”

Ln 371: “between individual bacterial taxa and”

Ln 377: “…or bacterial and methanogenic groups at the class level”

Ln 383, 393, 395: “were positively correlated”

Ln 386: “unclassified Firmicutes OTU was also positively…”

Ln 389: “suspect that the”

Ln 390: Use “absence of” or “lack of” rather than “missing” here.

Ln 397: “has previously been…”

Ln 404: “is strongly correlated”

Ln 407: Remove “observed”

Ln 416: “observed that the”

Ln 418: “as evidenced”

Ln 424: “matter”

Ln 429: “that there are”

6. PLOS authors have the option to publish the peer review history of their article (what does this mean?). If published, this will include your full peer review and any attached files.

Reviewer #1: No

Reviewer #2: **Yes: **Pramod Pandey

Reviewer #3: No

---

## [Author Response · Author response to Decision Letter 0]

24 May 2021

Response to Reviewers

We thank all reviewers for their time and thoughtful suggestions. We addressed each comment below and all line numbers in the responses are referring to the line numbers in the revised manuscript with tracked changes. These suggestions have helped to clarify and strengthen the manuscript, and we are grateful for the time and consideration of these reviews.

Reviewer #1: The authors compared physical and chemical properties of different types of swine manure pit samples (no-foam, foam, and crust), and made associations of these with the bacterial and methanogen community structure, and methane production rate. The work is interesting and generally well-performed, and it does shed light on some of the differences in these sample types. Despite these advances, the reviewer feels that the authors could have extended their observations by delving a bit more into the characteristics of the samples, particularly with regard to the SCFA and LCFA (see Specific comments below), as both types of compounds are major substrates or intermediates and their metabolic fates have a strong influence on methane production and, presumably, microbial community structure.

A second issue regards the pH of the samples, and how it might relate to SCFA concentration and to methane production. It appears from L66 and Supplemental Figure S3 (S3 Fig) that pH was measured, and there was a slight correlation of pH with some of the other measured characteristics (particularly nutrients), especially in the no-foam manures, but little correlation of pH with SCFA. This should perhaps be pointed out somewhere in the text. It would also be useful to indicate the range of pH values encountered, and whether they differed significantly among the manure types. As the aceticlastic methanogens tend to be more sensitive to acidic conditions than are the hydrogenotrophic methanogens, could pH explain the general absence of the former group in these samples?

Response: The reviewer made many helpful suggestions, and they are specifically addressed below. To address the reviewer’s comment on manure pH, the average pH was 8.17, 8.24, and 8.21 in no-foam, crust forming, and foaming manure, respectively. This slightly alkaline pH could be contributing to the absent of acetoclastic methanogens. This point has been added to the revised manuscript (L325-346). 

Specific comments:

L50: The authors bring up the tendency of foaming to cause a potential for explosion, but in the Results we learn that the foaming samples showed lower rates of methane production than did the other samples. Presumably the foaming causes methane accumulation even if it is produced in lesser amounts. This should be mentioned in the Discussion.

Response: Our results are consistent with the tendency of foaming to cause a potential for explosion. In Table 1, foaming manure had significantly higher rates of methane production. We double checked the manuscript to ensure that results stated that foaming samples was associated with higher methane production consistently. 

L64-67: The methods description is inadequate. Reference 4 does not provide analytical methods, and no methods are described for LCFA and SCFA analysis.

Response: We thank the reviewer for pointing it out. The correct reference with manure sampling and analyses has been added (L80). 

L115, L126: How were these sequence numbers (10k for bacteria, 4k for methanogens) selected, as opposed to normalizing the number of reads selected to number of reads in the sample with the fewest reads?

Response: The sequence number cut offs were evaluated based on Good’s coverage and total sequencing depth. Rarefaction has been argued as not suitable for microbial compositional analysis because it can introduce more biases (McMurdie and Holmes, 2014; Willis, 2019). In general, it is much more important to remove samples that were inadequately sequenced and thus may bias representation due to low sequencing coverage. Because the swine manure bacterial community is more diverse than the methanogen community, we set the Good’s coverage at 97% for bacterial samples and 99% for methanogen samples, which reflects 10k and 4k sequences, respectively. We have also updated this in the manuscript for clarification (L139-144, L155-157). 

McMurdie PJ, Holmes S: Waste not, want not: why rarefying microbiome data is inadmissible . PLoS Comput Biol. 2014, 10: 1003531-10.1371/journal.pcbi.1003531.

Willis, A. D. (2019). Rarefaction, alpha diversity, and statistics. Front. Microbiol. 10:2407. doi: 10.3389/fmicb.2019.02407

L158-161: It would be very useful to be more specific regarding these sample types, perhaps by including photographs (even if included in Supplementary data).

Response: Manure pictures with specific surface texture are shown below. They have been added to the Supplementary document as S2 Fig. Additional description to the sample types have also been added on L82-84. 

The picture (S2 Fig) illustrates the different surface textures after short term lab incubation. From left to right they are: No foam, No foam, Foam, Foam, Crust, Foam.

L181-183: Another important difference between SBM and DDGS is the amount and type of LCFAs, which should have some bearing in discussion of the importance of LCFA and its metabolism in the three sample types.

Response: We thank the reviewer for the suggestion. We have added this to the discussion (L281-286). 

L209-211: This seems like a superficial distinction, as the “physical interactions” must have some sort of underlying chemical basis (e.g., surface tension is probably governed by the relative contributions and hydrophobic/hydrophilic behavior of proteins, lipids and polysaccharides).

Response: We agree that the term “physical interactions” is not descriptive. The sentence has been updated to “These observations suggest that the manure surface texture change from no-foam to foam is associated with a shift from the dominance of correlations among physical properties to correlations among chemical properties, with crusts as an intermediate.” (L254-257). 

L228-254: In this section the authors discuss SCFA and acetic acid in the samples without providing any data on the amounts and distributions (i.e., different chain lengths) of these acids. Such information would be very useful in interpreting potential bottlenecks in LCFA degradation to SCFA, and SCFA catabolism to methane.

Response: We thank the reviewer for the good suggestion. The information about SCFA distribution has been added to the revised manuscript (L289-L292, L318-L321). 

L256: This is an oversimplification. Perhaps the authors should add “under standard conditions”. The process of LCFA degradation to SCFA typically operates near equilibrium, so that it is quite sensitive to the relative concentrations of substrate and product. But it does not typically require input of energy, and the authors seem to suggest this in the following sentences, in which they state that methanogenesis provides thermodynamic displacement via H2 consumption, allowing SCFA catabolism to occur.

Response: This has been updated in the revised manuscript as suggested by the reviewer (L347). 

L320-321: It is not clear to the reviewer how the individual slope values for the three types of manures can be compared to yield statistically significant differences, as there is only a single slope value for each sample type.

Response: The statistical significance of the community stability slopes was calculated using bootstraping, which repeatedly sample slopes based on different points within a group at random.

L349-360: These are interesting observations, but they raise the question of where the acetic acid goes. Working back to comments to L228-254 above, is it possible that some members of the bacterial community might be engaged in chain elongation reactions, in which acetate is converted to longer SCFA such as butyrate and caproate? Chain elongation has been widely observed in waste pits and anaerobic reactors in which aceticlastic methanogenesis is limited (for example, see Zhu et al., DOI: 10.1038/srep14360, or Angenent et al., DOI: 10.1021/acs.est.5b04847).

Response: These are very good questions. Unfortunately, we did not measure medium chain fatty acids for this study. However, they are worth investigating for future studies. We have added the reviewer’s suggestions to the paper on L354-357. 

L398-401: What is meant here by the vague statement regarding “the different potential involvement of bacteria or methanogens”?

Response: This sentence has been updated to clarify “the different potential involvement of bacteria or methanogens” as “The synergistic interactions between bacteria and methanogens could explain...” (L515-516).

L421-424: The reviewer is not convinced by this statement. Why can’t the accumulation of SCFA be due to impaired catabolism of SCFA, rather than invoking a greater production of SCFA from other feed components?

Response: We agree with the reviewer. This has been updated in the text on L545-546. 

S2 Table: Please define (in a footnote) the term “integrator”.

Response: This has been added in the revised Supplementary Information L83.

Minor edits:

L45-46: Delete “However,” and use rest of sentence as the first sentence of the next paragraph.

Response: Updated on L48.

L50: Delete “conditions of”.

Response: Updated on L52.

L185: Change “low” to “poorly”.

Response: Updated on L217.

L229: Here and elsewhere in the manuscript, change “SCFA/acetic acid” to just “SCFA”, because (as pointed out in L233-234) acetic acid is one of the SCFA.

Response: This has been updated throughout the manuscript.

L277: Change “between” to “among”.

Response: Updated on L377.

L329: Change “genera” to “genus”.

Response: Updated on L432.

L347: Change “that the bacterial or methanogen groups were the most abundant in” to “in which the bacterial or methanogen groups were the most abundant”.

Response: Updated on L455.

Reviewer #2: Manuscript entitled "Microbial assemblages and methanogenesis pathways impact on methane production and foaming in manure deep-pit storages" can be considered as high importance. This manuscript provides relatively new and unique dataset to tackle the human, animal and environmental risks associated with swine industry. The study provided a relatively large dataset to support the finding. Manuscript is well written and deserve publications.

Few things, however, authors are suggested to consider: 

Response: We thank the reviewer for the comments and suggestions. The specific updates and responses are below.

1) provide the details of farm (herd size; pit size; and existing manure management in farms where samples were collected); 

Response: We added S1 Table in the supplementary material to provide the additional farm information.

2) Figure 1 requires further descriptions in Figure captions (currently not clear); 

Response: We have updated the caption of Figure 1 (L203-205). 

3) method section requires further clarifications improvement in terms of sampling protocol, and the strategies implemented in sample collection; 

Response: We thank the reviewer for pointing this out. We updated the citation and detailed sampling protocols and strategies are described in the cited article (L80). 

4) before ending results and discussion section, authors are suggested to provide major findings quantitatively, and limitations of the study; 

Response: We thank the reviewer for the suggestion. We listed out our major findings on L519-535. Due to the descriptive nature of our study, many of the key findings cannot be expressed quantitatively, and after deliberation, felt that adding numbers the findings would interfere with the coherence of the summary. We added the limitations of the study on L546-548. 

and 5) discussion on the composition of gas formation, ammonia, and microbial community identified in Figure 5,6.7. 

Response: We thank the reviewer for the suggestions. Unfortunately, we do not have the information on gas compositions. We did not specifically discuss the negative correlation observed between ammonia and Gamma-Proteobacteria in the crust-forming manure because it deviates from the main findings of the study. Similarly, ammonia strongly correlated with potassium and TKN in all 3 manure types (S3 Fig), which were not unique to a specific foam type, and we do not know how they contributed to our overall findings. 

Any pathogens, in addition to methanogens were identified?

Response: The reviewer raised a very good question. We used microbial 16S rRNA gene based sequencing method to identify bacteria in this study. While this method can identify a wide range of bacterial genera and sometimes species, it does not detect the specific virulence or pathogenicity. One may be able to infer the pathogenicity of a species based on its literature presence, however, it is not reliable. We considered this discussion outside the scope of the current study. 

Further, in statistics table, authors are suggested to provide the sample numbers used for the analysis. In terms of feed characterization, not sure if the feed of individual farms has been characterized or the feed from mills which supplies the feed has been characterized. Each pit will have certain holding capacity, and the retention time will be having impacts of microbial community. Therefore, the conditions of pits (mixing, retention time, temperature, flushing frequency, and the frequencies of disloading of those pits are important consideration while describing the shift in microbial community under anaerobic conditions.

Response: The reviewer made many good suggestions. We added sample sizes to Table 1 (L241-242). Feed from individual farms were characterized and we also added S1 Table to address the reviewer’s question regarding the farms. We do not have all information the reviewer asked, but we did recognize the significant impact of pit conditions and hence used the core microbial communities of each manure type to address the major differences. We clarified this on L425-L427. 

Reviewer #3: This manuscripts describes the bacterial and methanogenic communities of manure collected from 46 swine farms at several different time points. The authors visually classified the manure surface as either non-foaming, crust-forming or foaming at each sampling time.

Certain chemical properties of the manure were also assessed. The authors report that the swine diet influences the manure surface texture and that certain chemical and physical properties were correlated with the manure surface texture as well. The non-foaming and foaming manure bacterial and methanogenic microbial communities were most dissimilar from each other.

Response: We appreciate reviewer’s time and suggestions. The specific comments are addressed below. 

Major comments

The materials and methods for the manure sampling and chemical analysis appear to be missing. The superscript “4” is included at the end of the sentence on ln 65 but it is unclear if this is referring to REF 4 or something else.

Response: We thank the reviewer for pointing it out. The correct reference on the manure sampling and chemical analysis has been added (L80). 

Minor comments

In S2 Table should the P-values for Farm and Feed Mill for bacteria also be 0.001?

Response: S2 Table has been updated as S3 Table. The P-values for Farm and Feed Mill is 1.0 because Farm is used as blocking effect as stated in the footnote. We noticed the discrepancies in the methanogen PERMANOVA values and have updated them. 

Can the x-axis in Fig. 5 C and D be converted to percent relative abundance rather than the current scale which is not very intuitive.

Response: We thank the reviewer for the suggestion. We have updated Fig. 5 to change the x-axis to percent scale.

Ln 24: “an increased”

Response: updated on L24. 

Ln 26-27: “…rRNA and archaeal mcrA genes…”

Response: updated on L26. 

Ln 29: “and that the bacterial…”

Response: updated on L28. 

Ln 31: “previously” should be removed since they are still uncharacterized.

Response: updated on L30. 

Ln 33, 362: “relative abundance”

Response: updated on L33 and L443. 

Ln 50: I would remove “conditions of”

Response: updated on L52. 

Ln 81: “mcrA gene was”

Response: updated on L96. 

Ln 111-113: What database was used for 16S rRNA gene classification?

Response: We used RDP 16S rRNA database. This information has been added on L137.

Ln 121: “chimeras”

Response: updated on L150. 

Ln 180: What is meant by “more significant” here? A greater concentration of DDGS? Also, what does the superscript “19” refer to here?

Response: The sentence and the reference have been updated to “... the proportion of distiller’s dried grains with soluble (DDGS) was significantly higher in feed given to swine from foam and crust manure...” (L212-213 and L215).

Ln 182: “NDF” should be first defined here rather than on ln 185.

Response: updated on L215. 

Ln 205, 407: “were correlated”

Response: This sentence has been updated to “... strong correlations among chemical properties were observed in foaming manures. Specifically the correlations among...” (L250-L253). 

Ln 228: The use of “SCFA/acetic acid” here and elsewhere is unclear to the review. Was acetic acid the only SCFA measured?

Response: Acetic acid was the largest component of SCFA measured. We updated the information on L289-L292. The use of “SCFA/acetic acid” has also be changed to “SCFA” throughout the paper. 

Ln 275: Remove “occurrence of”

Response: updated on L374. 

Ln 338: “of shared and unique A) bacterial and B) methanogenic OTUs of no-foam…”

Response: updated on L441.

Ln 340: “most relatively abundant”

Response: updated on L443.

Ln 342: “that differed”; “are shown”

Response: updated on L450-451.

Ln 346: “relatively abundant”

Response: updated on L455.

Ln 361: “taxa”

Response: updated on L469.

Ln 364: “was positively correlated”; “Bacilli (Lactobacillus) OTU and…”

Response: updated on L471 and L472.

Ln 366: I would remove “members of”

Response: updated on L478.

Ln 367-368: “…content was also negatively”

Response: updated on L480.

Ln 371: “between individual bacterial taxa and”

Response: updated on L484.

Ln 377: “…or bacterial and methanogenic groups at the class level”

Response: updated on L490.

Ln 383, 393, 395: “were positively correlated”

Response: updated on L496, 511, and 513.

Ln 386: “unclassified Firmicutes OTU was also positively…”

Response: updated on L499.

Ln 389: “suspect that the”

Response: updated on L507.

Ln 390: Use “absence of” or “lack of” rather than “missing” here.

Response: updated on L508.

Ln 397: “has previously been…”

Response: updated on L515.

Ln 404: “is strongly correlated”

Response: updated on L522.

Ln 407: Remove “observed”

Response: updated on L525.

Ln 416: “observed that the”

Response: updated on L539.

Ln 418: “as evidenced”

Response: updated on L541.

Ln 424: “matter”

Response: updated on L547.

Ln 429: “that there are” 

Response: updated on L554.

---

## [Decision Letter · Decision Letter 1]

16 Jun 2021

PONE-D-21-06556R1

Microbial assemblages and methanogenesis pathways impact on methane production and foaming in manure deep-pit storages

PLOS ONE

Dear Dr. Yang,

Thank you for submitting your manuscript to PLOS ONE. After careful consideration, we feel that it has merit but does not fully meet PLOS ONE’s publication criteria as it currently stands. Therefore, we invite you to submit a revised version of the manuscript that addresses the points raised during the review process.

We look forward to receiving your revised manuscript.

Kind regards,

Alex V Chaves, PhD

Academic Editor

PLOS ONE

Journal Requirements:

Reviewers' comments:

Reviewer's Responses to Questions

**Comments to the Author**

1. If the authors have adequately addressed your comments raised in a previous round of review and you feel that this manuscript is now acceptable for publication, you may indicate that here to bypass the “Comments to the Author” section, enter your conflict of interest statement in the “Confidential to Editor” section, and submit your "Accept" recommendation.

Reviewer #1: (No Response)

Reviewer #2: All comments have been addressed

Reviewer #3: All comments have been addressed

2. Is the manuscript technically sound, and do the data support the conclusions?

Reviewer #1: Yes

Reviewer #2: Yes

Reviewer #3: Yes

3. Has the statistical analysis been performed appropriately and rigorously? 

Reviewer #1: Yes

Reviewer #2: Yes

Reviewer #3: Yes

4. Have the authors made all data underlying the findings in their manuscript fully available?

Reviewer #1: Yes

Reviewer #2: Yes

Reviewer #3: Yes

5. Is the manuscript presented in an intelligible fashion and written in standard English?

Reviewer #1: Yes

Reviewer #2: Yes

Reviewer #3: Yes

6. Review Comments to the Author

Reviewer #1: The authors have adequately addressed the reviewers comments. One additional question arises:

L386: This does not make sense to the reviewer. Hydrogenotrophic methanogens oxidize H2, and use the reducing equivalents to reduce CO2. How do they oxidize acetate?

Minor edits:

Title: Suggest deleting “on”.

L43 and L62: Please use SI units (meters, rather than ft).

L119, L131: Change “less” to “fewer”.

L256: Change “microorganism metabolism” to “microbial metabolism”.

L257: Change “many” to “most”.

L279-281: This is a little confusing as written. Suggest changing “Under standard anaerobic conditions, the breakdown of LCFA is carried out via acetogenesis [35,42]. This process is energy-consuming and non-spontaneous.”, to “Under anaerobic conditions, the breakdown of LCFA is carried out via acetogenesis [35, 42]. This process is endergonic and does not occur spontaneously under standard conditions.”

L298: Insert “, respectively” after “methanogens”.

L458: Change “fibers” to “fiber”.

Reviewer #2: Authors revised manuscript, and revised version of manuscripts responded answers to comments line by line, and issues raised have been resolved.

Reviewer #3: (No Response)

7. PLOS authors have the option to publish the peer review history of their article (what does this mean?). If published, this will include your full peer review and any attached files.

Reviewer #1: No

Reviewer #2: **Yes: **Pramod Pandey

Reviewer #3: No

---

## [Author Response · Author response to Decision Letter 1]

1 Jul 2021

Reviewer #1: The authors have adequately addressed the reviewers comments. One additional question arises:

L386: This does not make sense to the reviewer. Hydrogenotrophic methanogens oxidize H2, and use the reducing equivalents to reduce CO2. How do they oxidize acetate?

Response: We thank the reviewer for the question. Acetate oxidation in the absence of acetolastic methanogens is part of the overall anaerobic methane production pathway. However, acetate oxidation is carried out by microorganisms other than hydrogenotrophic methanogens. We rephrased the sentence to clear the confusion. 

Minor edits:

Response: We thank the reviewer for the suggestions. The edits below have been incorporated. 

Title: Suggest deleting “on”.

L43 and L62: Please use SI units (meters, rather than ft).

L119, L131: Change “less” to “fewer”.

L256: Change “microorganism metabolism” to “microbial metabolism”.

L257: Change “many” to “most”.

L279-281: This is a little confusing as written. Suggest changing “Under standard anaerobic conditions, the breakdown of LCFA is carried out via acetogenesis [35,42]. This process is energy-consuming and non-spontaneous.”, to “Under anaerobic conditions, the breakdown of LCFA is carried out via acetogenesis [35, 42]. This process is endergonic and does not occur spontaneously under standard conditions.”

L298: Insert “, respectively” after “methanogens”.

L458: Change “fibers” to “fiber”.

Reviewer #2: Authors revised manuscript, and revised version of manuscripts responded answers to comments line by line, and issues raised have been resolved.

Reviewer #3: (No Response)

---

## [Editor Report · Decision Letter 2]

2 Jul 2021

Microbial assemblages and methanogenesis pathways impact methane production and foaming in manure deep-pit storages

PONE-D-21-06556R2

Dear Dr. Yang,

We’re pleased to inform you that your manuscript has been judged scientifically suitable for publication and will be formally accepted for publication once it meets all outstanding technical requirements.

Kind regards,

Alex V Chaves, PhD

Academic Editor

PLOS ONE

---

## [Editor Report · Acceptance letter]

22 Jul 2021

PONE-D-21-06556R2 

Microbial assemblages and methanogenesis pathways impact methane production and foaming in manure deep-pit storages 

Dear Dr. Yang:

I'm pleased to inform you that your manuscript has been deemed suitable for publication in PLOS ONE. Congratulations! Your manuscript is now with our production department. 

Kind regards, 

on behalf of

Prof Alex V Chaves 

Academic Editor

PLOS ONE